# RoCCo: Rotation-Augmented Clustering-based Low-rank Approximation for Compressing Large Language Models

## Abstract

The immense size and computational cost of Large Language Models (LLMs) present significant barriers to their widespread deployment. Low-Rank Approximation (LRA) offers a promising, hardware-friendly solution by factorizing large weight matrices into more compact forms. A key insight is that the accuracy of this factorization can be significantly enhanced by first applying a geometric transformation to the model's weights. In this work, we introduce RoCCo (Rotation-augmented Clustering for Compression), a novel LRA framework that uses clustering to factorize weight matrices. We first apply an orthogonal transform to shape restructure the weight geometry to be more suitable to clustering. We then apply a group-wise clustering algorithm to the transformed weights to achieve a precise approximation. Furthermore, we demonstrate that this factorized representation enables a novel clustered attention mechanism, which reduces the algorithmic complexity of inference by performing attention computations directly in the compressed domain. Through experiments on the LLaMA and OPT model families, we show that RoCCo can compress models by 75% while retaining over 96% of the original zero-shot accuracy on LLaMA2-13B achieving a competitive compression-accuracy trade-off.

## 1 Introduction

Large Language Models (LLMs) have become foundational across a vast area of applications, demonstrating remarkable performances in complex reasoning and generation tasks. However, this success is built on top of immense scale, with models containing billions of parameters that demand extensive computational and memory resources. The resulting costs create a significant barrier to their widespread deployment, making inference expensive and hindering their use in resource-constrained environments. Consequently, model compression has become a critical field of research aimed at making these powerful models more accessible and efficient.

While element-wise compression methods like scalar quantization (Lin et al., 2024b; Zhao et al., 2024; Frantar et al., 2023) have been heavily explored, our work focuses on the structural path of Low-Rank Approximation (LRA) (Yuan et al., 2023; Tian et al., 2025; Chen et al., 2021; Wang et al., 2025), specifically weight clustering to achieve a low-rank factorization of model weights, having the advantage of capturing and preserving the high-dimensional structure. This approach decomposes a weight matrix into a small set of learned centroids and corresponding matrix of integer indices, producing a hardware-friendly dense representation. However, the primary challenge for finding an accurate factorization is the presence of vector-wise outliers: a small number of out-of-distribution weight vectors that greatly influence the clustering process. These vectors stretch the geometric space, pulling cluster centroids away from dense regions and leading to a poor low-rank approximation for the majority of the in-distribution data.

A key insight from recent work is the principle of computational invariance, which allows for the application of orthogonal transformations to restructure a model's internal representations. This principle has been successfully leveraged to enable various compression goals, from reducing a model's embedding dimension (Ashkboos et al., 2024a) to improving uniform quantization perfor-

mance (Ashkboos et al., 2024b; Liu et al., 2025). However, its potential to improve the performance of clustering-based low-rank factorization remains a comparatively under-explored area.

In this work, we develop a methodology to address the vector-wise outliers that hinder the performance of low-rank approximation. We demonstrate that the principle of computational invariance can be adapted to this distinct problem by applying a random Hadamard transform to restructure the high-dimensional geometry of the weight vectors themselves. This transformation mitigates the influence of high-leverage vector outliers, creating a distribution that is significantly more amenable to being factorized by a clustering algorithm. This restructuring step allows us to find a more accurate low-rank approximation, enabling robust performance at high compression ratios.

Our main contributions are as follows:

1. We identify and quantify the effect of out-of-distribution outlier vectors in LLM weight matrices, showing they are a primary obstacle to effective low-rank approximation via weight clustering.

2. We propose a novel, low-rank approximation based compression method that uses a random Hadamard transform to restructure the weight geometry, mitigating the influence of outliers and making the weights significantly more amenable to clustering.

3. We show that our low-rank factorization approach is orthogonal to element-wise methods like quantization and can be combined to achieve further gains in efficiency and higher compression ratios.

4. We introduce a novel clustered attention mechanism that leverages our factorized representation to perform attention computation directly on the cluster centroids, offering an algorithmic speedup.

5. We demonstrate through extensive experiments that RoCCo achieves a state-of-the-art compression-accuracy trade-off. RoCCo is able to compress models by up to 75% while retaining over 96% of the original zero-shot accuracy on LLaMA2-13B, outperforming existing LLM low-rank approximation works.

## 2 BACKGROUND AND RELATED WORK

### 2.1 LOW-RANK APPROXIMATION OF LLM

Low-rank approximation (LRA) is a promising structural compression approach that aims to reduce model size by replacing large weight matrices with a more compact representation, often through factorization. Unlike element-wise methods that can produce sparse matrix representations, LRA-based methods typically result in smaller, dense matrices, which have high efficiency on modern hardware like GPUs. This approach has the advantage of capturing and preserving the high-dimensional structure of the weight matrices. The primary LRA techniques in recent literature can be categorized as follows:

**SVD-based**: A classic approach to LRA is through Singular Value Decomposition (SVD). Methods like SVD-LLM (Wang et al., 2025) and ASVD (Yuan et al., 2023) factorize a weight matrix into its constituent singular vectors and values, and then achieve compression by truncating the components corresponding to the smallest singular values. While this theoretically grounded, the performance of the approximation can degrade significantly at higher compression ratios.

**PCA-based**: Another family of methods uses Principal Component Analysis (PCA) on the weights or activations directly to identify a low-rank subspace and operate within this more compact subspace. SliceGPT (Ashkboos et al., 2024a), for example, uses PCA to find the principal components of the hidden states and then "slices" off the dimensions corresponding to the least important components, effectively reducing the model's embedding dimension. Similarly, FLAT-LLM (Tian et al., 2025) uses a fine-grained, head-wise PCA to achieve a more targeted low-rank transformation. ModeGPT (Lin et al., 2024a)

While effective, these LRA methods operate on the fixed geometry of the pre-trained weights. However, this geometry may not be optimal for LRA, particularly due to the presence of outliers (Lin et al., 2024b). This raises a critical question: rather than approximating the existing weights, can we first transform them into a structure that is inherently more amenable to a low-rank approximation?

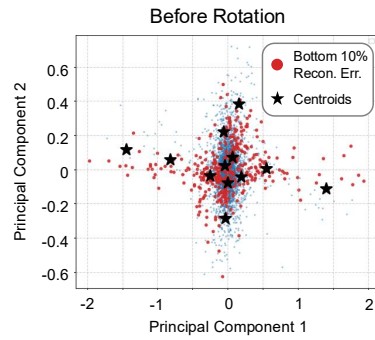 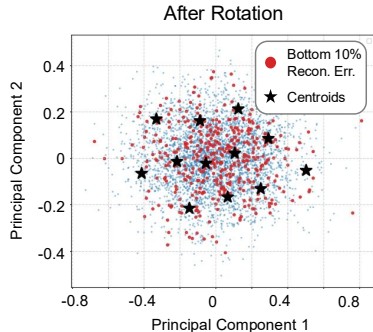

Figure 1: The effect of the orthogonal transform (RHT) on the weight geometry of a LLaMA2 weight layer (layer[5].self_attn.q_proj). The plots show a 2D PCA projection of the weight vectors, vectors with low 10% reconstruction errors, and 12 K-means centroids. (Left) The original distribution is characterized by a dense core and several out-of-distribution outlier vectors. These outlier vectors tend to have lower reconstruction errors. (Right) After the transformation, the distribution becomes more homogeneous, and the influence of the original outliers is mitigated.

## 2.2 COMPUTATIONALLY INVARIANT TRANSFORMATIONS

A recent advance in LLM compression is the principle of computational invariance, which states that an orthogonal transformation can be applied within the network without changing the model's final output, provided it is properly reversed. This allows for the restructuring of weights and activations into a more compression-friendly format while preserving the model's function. This invariance is possible because an orthogonal matrix $Q^T Q = QQ^T = I$ can be absorbed into adjacent weight matrices, enabling a powerful, lossless manipulation of the network's internal geometry

Specifically, if the output weight matrix of one block is $W_{out}$ and the input weight matrix of the subsequent block is $W_{in}$, the transformation is absorbed by modifying the weights as follows: the output matrix is post-multiplied by $Q$ ($W_{out} \rightarrow W_{out}Q$) , while the subsequent input matrix is pre-multiplied by its inverse, $Q^T$ ($W_{in} \rightarrow Q^T W_{in}$). This ensures that the model's end-to-end computation remains unchanged ($A(W_{out}Q)(Q^T W_{in}) \rightarrow AW_{out}(QQ^T)W_{in} = AW_{out}W_{in}$). This principle has been successfully applied to restructure weights for various compression goals. This principle has been successfully leveraged for various compression goals: SliceGPT (Ashkboos et al., 2024a) uses it for structured pruning, methods like QuaRot (Ashkboos et al., 2024b) and SpinQuant (Liu et al., 2025) use it to improve element-wise quantization, and QuIP# (Tseng et al., 2024) applies it to enable vector quantization with a predefined lattice codebook.

## 3 PROPOSED METHOD

### 3.1 RESTRUCTURING WEIGHT GEOMETRY WITH ORTHOGONAL TRANSFORMS

A primary challenge in applying low-rank approximation to LLMs is the inherent geometry of their weight matrices. While the weights contain significant structural redundancy due to the structured computational graph of transformers (eg. Multi-head self-attention)(Vaswani et al., 2017), their distributions are often characterized by the existence of outliers. As shown in recent works (Lin et al., 2024b; Xiao et al., 2023), these outliers stretch the geometric space, making the overall distribution difficult to compress.

Recent works have successfully demonstrated that orthogonal transformations can mitigate this issue. Methods like QuaRot (Ashkboos et al., 2024b) and SpinQuant (Liu et al., 2025) use rotations to address element-wise outliers within activation and weight distributions. By applying a transform like a Hadamard matrix, they create a more homogeneous, Gaussian-like distribution of scalar values, which is ideal for low-bit uniform quantization.

Our work builds on this principle but re-frames the outlier problem for a different compression paradigm: clustering-based low-rank approximation. For this approach, the primary challenge is

not individual outlier values, but entire vector-wise outliers that distort the clustering process. As visualized through PCA in Figure 1, the weight vectors are mostly organized in a dense clump, but a small number of out-of-distribution (OOD) outlier vectors deviate significantly from this main distribution.

A key finding from our analysis is that many of the out-of-distribution (OOD) outlier vectors (geometrically distant from main distribution) are often found in the bottom 10% of vectors when ranked by individual reconstruction error (marked red in Figure 1). However, when applying clustering to the entire set of vectors, these OOD vectors act as powerful "gravitational centers," forcing the clustering algorithm to allocate centroids to these few distant points. This distorts the placement of the centroids, which are now less able to accurately represent the vast majority of the non-outlier vectors in the dense core of the distribution. It is the high reconstruction error of these "inlier" vectors (responsible for more than 95% of the entire reconstruction error) that is most important and that ultimately degrades the model's performance. Therefore, our goal is to apply an orthogonal rotation to restructure the weight geometry, mitigating the distorting influence of the outlier vectors to allow the centroids to form a more accurate representation of the main data distribution.

To achieve this restructure of the weight geometry for better clustering performance, we leverage a fixed random Hadamard transform (RHT) as shown by the right diagram of Figure 1. This choice is motivated by several advantages. First, the fast Walsh-Hadamard transform allows the rotation to be computed efficiently in $O(n \log n)$ time, instead of $O(n^2)$ of a regular transform in dimension $n$. Second, as an orthogonal matrix, it is a norm-preserving operation that rotates the weight vectors without distorting the relative distances between vectors. Finally, it has been shown to be an effective method by creating a more incoherent, or homogeneous distribution. This restructuring through RHT is a pre-processing step that does not alter the model's output but significantly improves the performance of the subsequent clustering-based compression method.

### 3.2 ORTHOGONAL TRANSFORM-BASED CLUSTERING

Inspired by the computationally invariant transformations in QuaRot (Ashkboos et al., 2024b), our proposed framework achieves a low-rank approximation of a given weight matrix through a three-stage process. This layer-wise method first regularizes the geometry of the weight distribution before learning a compact, clustered representation.

**Stage 1: Geometric restructuring via Hadamard transform.**

First, for a given weight matrix $\boldsymbol{W} \in \mathbb{R}^{d_{in} \times d_{out}}$, we apply a fixed, random Hadamard transform to regularize its structure. The transformed matrix is computed as $\boldsymbol{W}_{rot} = \boldsymbol{W}\boldsymbol{H}$, where $\boldsymbol{H}$ is a random Hadamard matrix of size $d_{out} \times d_{out}$. This preconditioning step mitigates the influence of high-magnitude outlier vectors, creating a more well-conditioned distribution for the subsequent clustering stage.

**Stage 2: Low-rank approximations via group-wise clustering**

The second stage performs the compression using a group-wise clustering algorithm, to split the weight matrix into smaller, more manageable parts. The objective is to independently cluster the columns within horizontal partitions of the weight matrix. First, the rotated weight matrix $\boldsymbol{W}_{rot} \in \mathbb{R}^{d_{in} \times d_{out}}$ is partitioned along its row dimension. The $d_{in}$ rows are divided into $G$ groups, where each group contains $k$ rows, such that $d_{in} = G \times k$. Where the partition factor $k$ is a hyperparameter. Smaller values of $k$ allow for a more fine-grained clustering at the cost of increased storage overhead for the additional cluster indices. This creates $G$ independent sub-matrices:

$$W_{rot} = \begin{bmatrix} W_{rot}^{(1)} \\ \vdots \\ W_{rot}^{(G)} \end{bmatrix} \tag{1}$$

where each sub-matrix $\boldsymbol{W}_{rot}^{(i)} \in \mathbb{R}^{k \times d_{out}}$ contains the $i$-th group of $k$ row.

We define our clustering process, $\phi(\cdot)$, as the function that maps each sub-matrix to its compressed representation. For each horizontal sub-matrix $\boldsymbol{W}_{rot}^{(i)}$, the algorithm partitions its $d_{out}$ columns

(each of row dimension $k$) into a dedicated set of $c$ centroids. This results in $G$ independent sets of centroids, $\boldsymbol{C}^{(i)} \in \mathbb{R}^{k \times c}$, and $G$ corresponding index vectors, $I^{(i)} \in \mathbb{Z}^{d_{out}}$.

The reconstruction process, $\phi^{-1}(\cdot)$, then uses these components to form the approximated weight matrix $\hat{W_{rot}}$. This reconstruction achieves a low-rank factorization of the original sub-matrix, given by:

$$\hat{\boldsymbol{W}_{rot}^{(i)}} = \phi^{-1}(\mathbf{C}^{(i)}, \mathbf{I}^{(i)}) = \mathbf{C}^{(i)} \cdot \boldsymbol{S}^{(i)} \tag{2}$$

where $S^{(i)}$ is a selection matrix constructed from the index vector $\mathbf{I}^{(i)}$.

The full approximated matrix is the vertical concatenation of these reconstructed sub-matrices. The primary benefit is a significant reduction in storage. The original matrix requires $d_{in} \times d_{out}$ values, while the compressed form before the reconstruction requires storing a total of $c \times d_{in}$ values for the centroids and $G \times d_{out} \times \lceil \log_2 c \rceil$ bits for the indices. For typical LLM configurations where the number of clusters $c/d_{out} < 0.5$ and the partition factor $k > 8$, the total storage cost of these indices is minimal, typically accounting for less than 2% of the original weight matrix size (see Appendix A.4).

**Stage 3: Data-aware centroid update.** The final stage calibrates the centroids obtained in Stage 2. We use a modified method based on GPTQ (Frantar et al., 2023) that updates the centroids to minimize the layer's output reconstruction error, using a small calibration set of data. The details on the clustering-based low-rank approximation are explained in detail in Appendix A.2.

### 3.3 Direct Computation on Clustered Weights

Beyond significant storage savings, our group-wise clustering method enables a more efficient computation path by avoiding the need to de-cluster, or reconstruct, the full weight matrix. Instead, matrix multiplications can be performed directly in the compressed, clustered domain, which reduces both the number of floating-point operations (FLOPS).

In a standard linear layer, the computation is a matrix multiplication between the input activations $\boldsymbol{X} \in \mathbb{R}^{n \times d_{in}}$ and the weight matrix $\boldsymbol{W} \in \mathbb{R}^{d_{in} \times d_{out}}$. Our method reformulates this computation into a two-step process:

Look-up Table Generation: For each of the $G$ groups, we first perform a much smaller matrix multiplication between the input activations $\boldsymbol{X}^{(i)}$ and the corresponding centroid matrix $\boldsymbol{C}^{(i)} \in \mathbb{R}^{k \times c}$. This creates $G$ small look-up tables, where each table contains the $c$ possible outputs for that cluster group.

$$\boldsymbol{X} = [\boldsymbol{X}^{(1)}, \boldsymbol{X}^{(2)}, \ldots, \boldsymbol{X}^{(G)}] \tag{3}$$

$$\boldsymbol{Y}^{(i)} = \boldsymbol{X}^{(i)} \cdot \mathbf{C}^{(i)} \tag{4}$$

Gather Operation: The final output matrix $\boldsymbol{Y} \in \mathbb{R}^{n \times d_{out}}$ is then constructed by efficiently gathering the appropriate values from these look-up table $\boldsymbol{Y}^{(i)}$ using the stored indices $\mathbf{I}^{(i)}$.

This reformulation avoids the costly reconstruction of the full $d_{in} \times d_{out}$ weight matrix. The computational cost is dominated by the initial look-up table generation, which is significantly cheaper than the original multiplication. Reducing the required computation from $O(nd_{in}d_{out})$ to $O(nd_{in}c)$ where $c \ll d_{out}$, decreasing linearly with the compression ratio $(1 - c/d_{out})$.

## 4 Experiments

### 4.1 Setup

We implement RoCCo using the QuaRot repository, which is built upon the HuggingFace(Wolf et al., 2019) and PyTorch framework(Paszke et al., 2019). For the main results presented in Tables 1, 2, 3, RoCCo is configured with a partition factor $k = 16$. We use a simple K-means algorithm for clustering, and the centroids are calibrated using 128 samples WikiText-2 (Merity et al., 2016) with a 2048 sequence length with a modified algorithm based on GPTQ(Frantar et al., 2023)(Appendix A.2). We evaluate our work on single NVIDIA A100 and NVIDIA H200 GPU across the OPT(Zhang et al., 2022), LLaMA1(Touvron et al., 2023a), and LLaMA2(Touvron et al., 2023b) model families, covering both language generation and zero-shot evaluation tasks.

Table 1: Comparison of WikiText-2 Perplexity (ppl) at various compression ratios across OPT and LLaMA2 model families. We compare our method RoCCo, against several LRA baselines.

| | | OPT | | | | | LLaMA2 | |
|---|---|---|---|---|---|---|---|---|
| Method | Ratio | 125M | 1.3B | 2.7B | 6.7B | 13B | 7B | 13B |
| Dense | - | 27.65 | 14.63 | 12.47 | 10.86 | 10.13 | 5.47 | 4.88 |
| DISP-LLM | 20% | 39.87 | 21.70 | 17.07 | 14.06 | - | 9.84 | 7.11 |
| SliceGPT | 20% | 34.26 | 16.43 | 13.73 | 11.48 | 10.66 | 6.64 | 5.81 |
| ProcrutesGPT | 20% | 36.08 | - | 13.95 | - | 10.67 | 6.54 | 5.71 |
| SVD-LLM | 20% | - | - | - | - | - | 7.84 | 7.37 |
| **RoCCo** | 20% | **27.91** | **14.93** | **12.68** | **11.00** | **10.22** | **5.49** | **4.94** |
| **RoCCo** | 50% | 31.19 | 16.11 | 12.93 | 11.12 | 10.29 | 5.56 | 4.99 |
| **RoCCo** | 75% | 32.44 | 16.79 | 13.32 | 11.32 | 10.45 | 5.71 | 5.04 |

Table 2: Zero-shot results for LLaMA2-7B, LLaMA2-13B models. SliceGPT, SVD-LLM, FLAT-LLM, ProcrutesGPT are evaluated with a compression ratio of 20%. RoCCo is evaluated with much higher compression ratio of 75%.

| Model | Method | Ratio | ARC-c | ARC-e | HellaS. | PIQA | WinoG. | Average |
|---|---|---|---|---|---|---|---|---|
| | Dense | 0% | 46.25 | 74.58 | 75.99 | 79.11 | 68.82 | 68.95 |
| | SliceGPT | 20% | 35.15 | 56.10 | 53.04 | 65.78 | 62.98 | 54.61 |
| | SVD-LLM | 20% | 35.84 | 68.31 | 58.57 | 71.49 | 65.27 | 59.90 |
| LLaMA2-7b | FLAT-LLM | 20% | 38.65 | 64.44 | 64.72 | 72.20 | 65.82 | 61.17 |
| | ProcrustesGPT | 20% | 41.98 | 68.35 | 69.72 | 73.94 | **67.40** | 64.28 |
| | RoCCo | 75% | **42.32** | **70.21** | **72.72** | **76.88** | 63.77 | **65.18** |
| | Dense | 0% | 49.23 | 77.53 | 79.36 | 80.52 | 72.30 | 71.79 |
| | SliceGPT | 20% | 39.51 | 62.92 | 56.98 | 67.25 | 67.64 | 58.86 |
| | SVD-LLM | 20% | 39.93 | 71.00 | 63.47 | 72.91 | 67.17 | 62.90 |
| LLaMA2-13b | FLAT-LLM | 20% | 46.25 | **75.59** | 73.36 | 75.84 | **72.06** | 68.62 |
| | ProcrustesGPT | 20% | 44.97 | 73.19 | 73.43 | 77.58 | 70.48 | 67.93 |
| | RoCCo | 75% | **47.61** | 73.91 | **76.86** | **78.89** | 68.82 | **69.22** |

## 4.2 ACCURACY RESULTS

**Language Generation Tasks**: We evaluate the performance of RoCCo on the language generation task using the WikiText-2 dataset (Merity et al., 2016). Table 1 shows a comparison of perplexity for the OPT and LLaMA2 model families against several state-of-the-art low-rank approximation (LRA) baselines.

At a 20% compression ratio, RoCCo demonstrates exceptional performance. Across all models, it consistently and significantly outperforms other LRA methods like DISP-LLM(Gao et al., 2024), SliceGPT(Ashkboos et al., 2024a), SVD-LLM(Wang et al., 2025), and ProcrustesGPT(Grishna et al., 2025). Notably, for the LLaMA2 models, RoCCo at 20% compression achieves a perplexity that remains comparable to that of the original dense model, which underscores the effectiveness of the approximation.

The primary advantage of our approach, is its robustness at high compression ratios. This high compression rate is possible because the selection matrix that maps the centroids to the original column dimensions, a large and sparse matrix of zeros and ones, does not need to be stored explicitly. Instead, it can be efficiently represented by a small vector of integer indices, where each index specifies which cluster group a column belongs to. As shown in the table, RoCCo maintains very strong performance even at 50% and 75% compression, a regime where other LRA based methods are not typically evaluated or often degrade significantly. For instance, on OPT-6.7B, RoCCo at 75% compression (11.32 ppl) outperforms the other baselines at 20% compression (14.06 ppl). This highlights that our geometric restructuring approach allows for high-ratio structural compression.

Table 3: An evaluation of combining RoCCo with GPTQ quantization. This hybrid approach significantly outperforms standard GPTQ, particularly in the 2-bit regime where GPTQ alone fails, demonstrating that RoCCo can be used as a method to compress models beyond what only quantization can do.

| Method | Effective bits | LLaMA1 | | LLaMA2 | |
| --- | --- | --- | --- | --- | --- |
| | | 7B | 13B | 7B | 13B |
| Dense | 16 | 5.68 | 5.09 | 5.47 | 4.88 |
| GPTQ | 4 | 6.10 | 5.36 | 6.09 | 5.16 |
| RoCCo (50%)+GPTQ(8b) | 4.34 | 5.91 | 5.23 | 5.72 | 5.04 |
| GPTQ | 2 | 17920 | 4127 | 6064 | 1960 |
| RoCCo (50%)+GPTQ(4b) | 2.34 | 12.57 | 10.81 | 11.52 | 10.20 |

**Zero-shot Tasks**: We further evaluate RoCCo's performance on a suite of five zero-shot commonsense reasoning tasks: ARC-challenge (Clark et al., 2018), ARC-easy (Clark et al., 2018), HellaSwag (Zellers et al., 2019), PIQA (Bisk et al., 2020), and WinoGrande (Sakaguchi et al., 2021). Table 2 presents a comparison against several state-of-the-art LRA baselines including SliceGPT (Ashkboos et al., 2024a), SVD-LLM (Wang et al., 2025), FLAT-LLM (Tian et al., 2025), and ProcrutesGPT (Grishna et al., 2025). All baselines are evaluated at a 20% compression ratio, while RoCCo is evaluated at a much higher 75% compression ratio.

For the LLaMA2-7B model, RoCCo at 75% compression achieves an average accuracy of 65.18%, which not only outperforms all other LRA methods at 20% compression but is also higher than the next best baseline FLAT-LLM by nearly a full percentage point. This shows that our method compressed by an additional 55% still retains more zero-shot capability.

This trend continues with the larger LLaMA2-13B model. RoCCo achieves an average score of 69.22%. Compared to the dense model's performance, our 75% compressed LLaMA2-13B model retains over 96% of the original zero-shot accuracy, demonstrating that our method is highly effective at preserving the downstream task performance of LLMs even under high compression.

**Joint Clustering & Quantization**: To demonstrate that our structural compression method is orthogonal to and compatible with conventional element-wise compression techniques, we evaluate a hybrid approach that combines RoCCo with a standard post-training quantization method, GPTQ (Frantar et al., 2023). Table 3 shows the results of applying GPTQ to our already-compressed models.

At approximately 4.34-bit precision (0.34 additional bits accounting for cluster indices), the combination of RoCCo (50%) with 8-bit GPTQ significantly outperforms the standard 4-bit GPTQ baseline. For the LLaMA2-13B model, our hybrid approach achieves a perplexity of 5.04, which is an improvement over GPTQ's 5.16 showing the complementary benefits of the two methods.

The most significant advantage, however, is observed in the extreme low-bit regime. As shown in the table, standard 2-bit GPTQ fails completely, resulting in an unusable model with extremely high perplexity. In contrast, our hybrid approach of RoCCo (50%) with 4-bit GPTQ, at a comparable effective bit of 2.34 bits, produces a stable and functional model with a reasonable perplexity.

## 4.3 ABLATION STUDIES

To validate our key design choices, we conduct a series of ablation studies. Our analysis confirms that our Hessian-based centroid calibration is a critical component, offering a clear improvement over a simple k-means baseline. We also demonstrate our performance with regards to the choice of the partition size, $k$. To further contextualize our contributions a detailed comparison with the state-of-the-art method QuIP# (Tseng et al., 2024) is provided to contextualize the trade-offs of our data-driven approach. Full results for all ablation studies can be found in the Appendix.

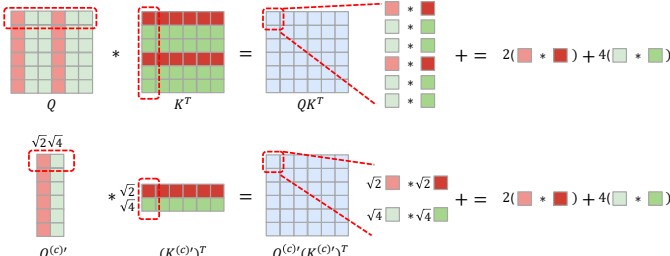

Figure 2: An illustration of the clustered attention mechanism. The same colored vectors are identical vectors of the same centroid. (Top) The standard $QK^T$ computation where the Query and Key matrices have a low-rank structure with matched QK clustering imposed. (Bottom) Our reformulation, which shows that the full dot product is mathematically equivalent to a matrix multiplication between smaller, scaled activation matrices $Q^{(c)'}$ and $K^{(c)'}$, reducing the computational complexity.

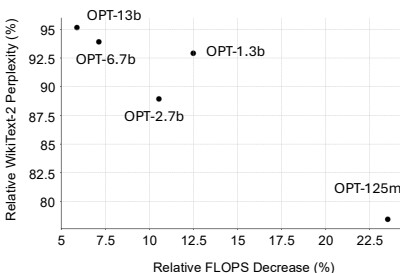

Figure 3: Performance trade-off for our clustered attention mechanism across the OPT model family. The y-axis shows the perplexity relative to the dense baseline (higher is better), while the x-axis shows the percentage decrease in computational FLOPs of the entire model.

## 5 EFFICIENT ATTENTION IN THE CLUSTERED DOMAIN

While RoCCo significantly reduces the model's memory requirements, it also enables a more efficient computation of the multi-head self-attention mechanism. By leveraging the clustered representation of the weight matrices, we can reformulate the attention score calculation to reduce its complexity.

In a standard multi-head self-attention layer, the input activations $\boldsymbol{X} \in \mathbb{R}^{N \times d_{model}}$ are projected into Query ($\boldsymbol{Q}$), Key ($\boldsymbol{K}$), and Value ($\boldsymbol{V}$) matrices using their respective weights. The scaled dot-product attention of a single head is then formulated as:

$$\text{Attention}(\boldsymbol{Q}, \boldsymbol{K}, \boldsymbol{V}) = \text{softmax}\left(\frac{\boldsymbol{Q}\boldsymbol{K}^T}{\sqrt{d_{head}}}\right)\boldsymbol{V} \tag{5}$$

The primary computational bottleneck in this operation is the $\boldsymbol{Q}\boldsymbol{K}^T$ matrix multiplication, which scales quadratically with the sequence length N. Our compressed representation avoids the need to compute this large matrix multiplication directly.

To alleviate this high computational requirement, we introduce a matched query-key clustering scheme. The core of this method is that for each attention head, the columns of the query weight matrix ($\boldsymbol{W}_q$) and the key weight matrix ($\boldsymbol{W}_k$) are clustered into $G$ clusters using the same cluster index assignments. Figure 2 shows a sample illustration of the clustered attention through matched query-key clustering.

This "matching" imposes a specific low-rank structure on the resulting Query ($\boldsymbol{Q}$) and Key ($\boldsymbol{K}$) activation matrices. All columns within a given group become identical copies of a corresponding centroid vector. This allows the activation matrices to be factorized as:

$$\boldsymbol{Q} = \boldsymbol{C}^{(Q)}\boldsymbol{S} \quad \text{and} \quad \boldsymbol{K} = \boldsymbol{C}^{(K)}\boldsymbol{S} \tag{6}$$

where $\boldsymbol{C}^{(Q)}, \boldsymbol{C}^{(K)} \in \mathbb{R}^{N \times G}$ are the matrices containing the effective activation values for each of the $G$ cluster groups. The matrix $\boldsymbol{S} \in \mathbb{R}^{G \times d_{head}}$ is a selection matrix, determined by the shared column clustering, that maps these group activations to the original head dimension.

This shared structure allows for a significant simplification of the $\boldsymbol{Q}\boldsymbol{K}^T$ matrix multiplication. The score between the $i$-th query token and the $j$-th key token can be shown to be a weighted sum of the products of their group activation values:

$$\boldsymbol{Q}\boldsymbol{K}_{i,j}^T = \sum_{g=1}^{G} |G^{(g)}| \cdot \boldsymbol{C}_{i,g}^{(Q)} \cdot \boldsymbol{C}_{j,g}^{(K)} \tag{7}$$

where $|G^{(g)}|$ is the size (number of column vectors) of the $g$-th cluster group.

Crucially, this entire $N \times N$ matrix multiplication can be rewritten. The sum of weighted dot products is equivalent to a single, much smaller matrix multiplication involving only scaled centroids. We define the scaled centroid matrices as:

$$\boldsymbol{Q}^{(c)'} = \boldsymbol{C}^{(Q)} \cdot \text{diag}(\sqrt{|G^{(1)}|}, \dots, \sqrt{|G^{(G)}|}) \in \mathbb{R}^{N \times G} \tag{8}$$

$$\boldsymbol{K}^{(c)'} = \boldsymbol{C}^{(K)} \cdot \text{diag}(\sqrt{|G^{(1)}|}, \dots, \sqrt{|G^{(G)}|}) \in \mathbb{R}^{N \times G} \tag{9}$$

Since this scaling is applied column-wise to the group activation matrices, it can be absorbed directly into the original weight matrices $(\boldsymbol{W}_Q, \boldsymbol{W}_K)$ prior to test time without any change to the final computation.

The full attention score matrix can now be computed efficiently as:

$$\boldsymbol{Q}\boldsymbol{K}^T = \boldsymbol{Q}^{(c)'}(\boldsymbol{K}^{(c)'})^{T1} \tag{10}$$

This reduces the complexity of the score calculation from $O(N^2 d_{head})$ to $O(N^2 G)$, where the number of cluster groups $G$ is significantly smaller than the head dimension $d_{head}$. This provides a direct and substantial algorithmic speedup without any look-up or gather operations.

The results of the clustered attention mechanism is presented in Figure 3. The results demonstrate a clear trade-off between computational savings and model performance for our clustered attention mechanism. Larger models tend to be more resilient to the clustered attention approximation's structure of matched QK clustering. For instance, OPT-13b retains over 95% of the original WikiText-2 perplexity while saving 5% of computation in the entire model. Smaller models, while more sensitive to the approximation of clustered attention, can achieve much more significant computational reductions. The OPT-125m model can reach a total FLOPS decrease of upto 22%. However, this does come at the cost of a higher drop in relative model performance.

## 6 CONCLUSIONS

In this work, we introduce RoCCo, a compression framework that combines geometric restructuring with clustering-based low-rank approximation. We demonstrate that pre-processing weights with a random Hadamard transform makes them significantly more suitable for clustering, enabling robust performance at high compression ratios where other structural methods often fail. Our experiments show that RoCCo achieves a strong trade-off between model size and performance compared to other low-rank approximation based compression techniques. Furthermore, we introduce two key benefits of our approach. First, our structural compression is orthogonal to element-wise methods, and we show that combining RoCCo with conventional quantization enables stable, low-bit representations where standard quantization collapses. Second, we proposed a novel clustered attention mechanism, which leverages our compressed format to reduce the algorithmic complexity of inference. RoCCo presents a promising and efficient path toward developing smaller, faster Large Language Models.

---

[1] Architectures that apply non-linear operations like Rotary Position Embeddings to the Queries and Keys before the dot product would require a different formulation.

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

# A  APPENDIX

## A.1  OUTLIER PROJECTIONS FOR DIFFERENT WEIGHTS

Figure 4 presents an ablation study on the effect of vector outliers for several attention projection matrices in the LLaMA2-7B model. The ablation study confirms that the out-of-distribution vectors often account for the lower reconstruction error, harming the clustering performance by increasing errors in the dense region. After applying our rotational transform, the influence of these outlier vectors is effectively mitigated as they are homogenized into the main distribution.

## A.2  CENTROID CALIBRATION WITH HESSIAN-BASED ERROR COMPENSATION

The initial centroids generated by k-means minimize the simple Euclidean distance, but this does not guarantee minimal error in the layer's output due to the input differences. Therefore, the final stage of our method refines the centroids by directly minimizing the layer's output reconstruction error, using an iterative, data-aware algorithm inspired by GPTQ (Frantar et al., 2023).

The overall objective is to find a set of centroids $C$ that minimizes the squared error of the layer's output, using a small calibration set of input activations $X$:

$$\underset{C}{\mathrm{argmin}}||XW_{rot} - X\hat{W}_{rot}||_2^2 \tag{11}$$

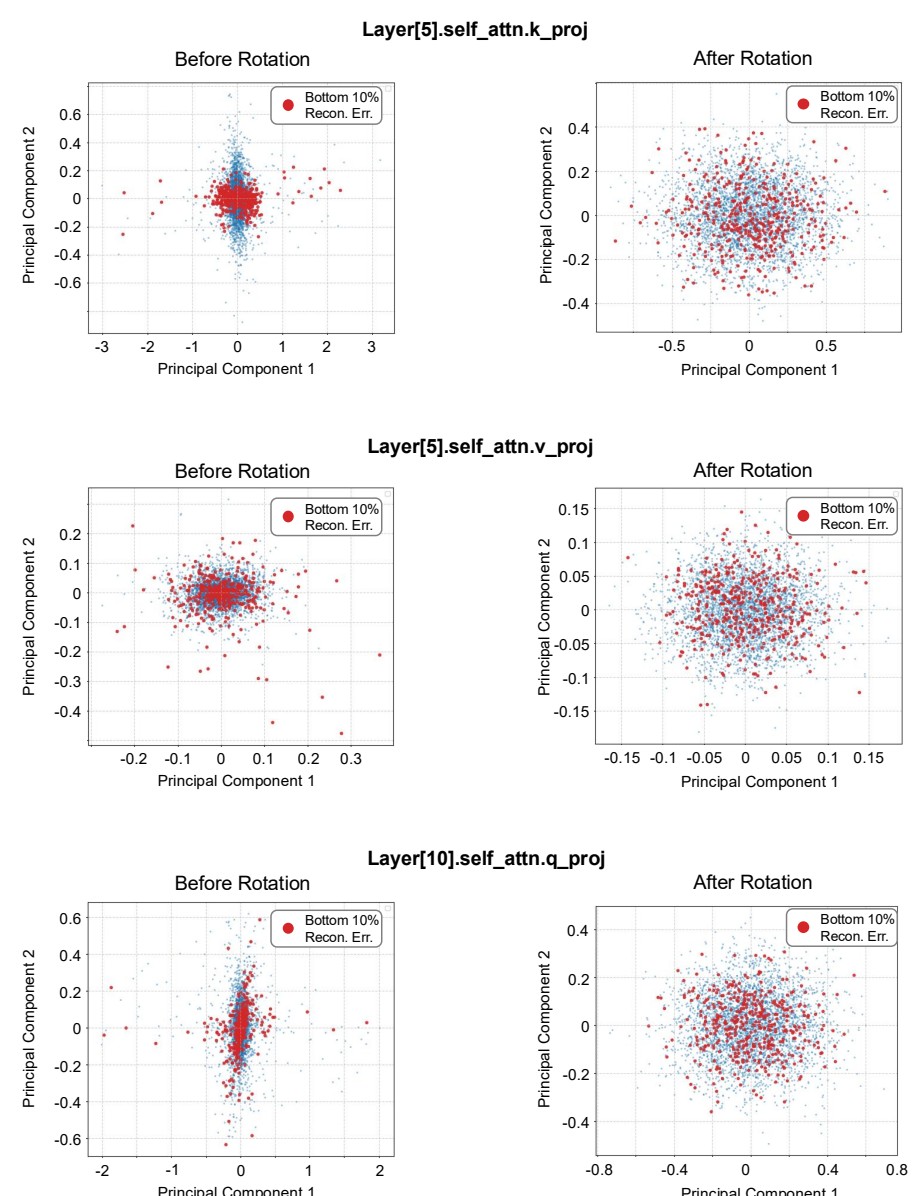

Figure 4: Projected visualization of weight matrices and its bottom 10% reconstruction error vectors before and after rotation.

where $\hat{W}_{rot}$ is the weight matrix reconstructed from the refined centroids. This objective can be simplified to minimizing the error with respect to a proxy Hessian.

Our algorithm, detailed in Algorithm 1&2, solves this objective for each of the $G$ sub-matrices independently. For a given weight matrix, the algorithm processes its columns one by one in a greedy, iterative fashion. In each step, it assigns a column to the corresponding centroid value and then propagates the resulting error to all remaining, not-yet-processed columns. This "assign-and-update" procedure continues until all columns have been processed, ensuring that the final set of centroids is refined to minimize the layer's functional output error.

---

**Algorithm 1** Cluster Rotated Weight $\boldsymbol{W}_{rot}$

---

**Require:** Rotated Weight $\boldsymbol{W}_{rot}$, Cluster Number $c$, Partition Factor $k$.
1: $\boldsymbol{W}' \leftarrow \boldsymbol{W}_{rot}$
2: $G \leftarrow d_{row}/k$                ▷ Determine group number based on partition factor
3: **for** $i$ in $G$ **do**
4:      $\boldsymbol{W}_{rot}^{(i)} \leftarrow \boldsymbol{W}'_{i\cdot k:(i+1)\cdot k-1,:}$             ▷ Split rotated weight into groups
5:      $\mathbf{C}^{(i)}, \mathbf{I}^{(i)} \leftarrow \text{KMeans}(\text{columns of } \boldsymbol{W}_{rot}^{(i)}, c)$   ▷ Cluster and obtain initial centroids & indices
6: **end for**
7: **return** $\mathbf{C}^{(i)}, \mathbf{I}^{(i)}$                            ▷ Return the final result

---

**Algorithm 2** Centroid Calibration for a Weight Matrix $\boldsymbol{W}_{rot}$

---

**Require:** Weight $\boldsymbol{W}_{rot}$, Hessian $\mathbf{H}$, Initial centroid $\mathbf{C}$, Cluster indices $\mathbf{I}$
1: $\mathbf{W}' \leftarrow (\boldsymbol{W}_{rot})^T$           ▷ Initialize a temporary matrix to hold the updatable weights
2: $\mathbf{H}^{-1} \leftarrow \text{invert}(\mathbf{H})$
3: **for** $j \leftarrow 1$ to $d_{col}$ **do**
4:      $\mathbf{c}_{assigned} \leftarrow \mathbf{C}_{:,\mathbf{I}_j}$                 ▷ Get the current centroid for column $j$
5:      $\boldsymbol{\Delta}_{:,j} \leftarrow (\mathbf{W}'_{:,j} - \mathbf{c}_{assigned})/\mathbf{H}^{-1}_{j,j}$       ▷ Calculate the error for the current column
6:      $\mathbf{W}'_{:,j+1:d_{col}} \leftarrow \mathbf{W}'_{:,j+1:d_{col}} - \boldsymbol{\Delta}_{:,j} \cdot \mathbf{H}^{-1}_{j,j+1:d_{col}}$ ▷ Update remaining columns with the error
7:      $\mathbf{C} \leftarrow$ recompute centroids from $\mathbf{W}'$ using the fixed indices $\mathbf{I}$         ▷ Update centroid
8: **end for**
9: **return** $\mathbf{C}, \mathbf{I}$

---

## A.3 CENTROID CALIBRATION RESULTS

Table 4: Comparison of RoCCo with/without centroid calibration process

| Method | LLaMA1 | | LLaMA2 | |
|---|---|---|---|---|
| | 7B | 13B | 7B | 13B |
| Dense | 5.68 | 5.09 | 5.47 | 4.88 |
| RoCCo (20%) w/o calibration | 35.46 | 28.27 | 31.10 | 27.36 |
| RoCCo (20%) w/ calibration | 5.73 | 5.14 | 5.49 | 4.94 |
| RoCCo (50%) w/o calibration | 60.92 | 48.33 | 55.42 | 48.97 |
| RoCCo (50%) w/ calibration | 5.77 | 5.18 | 5.56 | 4.99 |

Table 4 presents the ablation study on the effect of the centroid calibration process. We compare the full RoCCo method against a baseline version that omits the final centroid calibration stage.

The results show that the centroid calibration step is critical to the success of our framework. The baseline method, which relies solely on the initial k-means centroids, has performance degradations at both 20% and 50% compression ratios, resulting in perplexity scores that are worse than the dense model. This indicates that a simple Euclidean-distance-based clustering is insufficient to preserve the output of the layer.

In contrast, our full RoCCo method, which includes the data-aware centroid calibration, yields a dramatic improvement. At a 20% compression ratio, the calibration step reduces the perplexity on the LLaMA2-7B model from 31.10 to 5.49, which is nearly identical to the dense model's performance. This demonstrates that our Hessian-based refinement is highly effective at compensating for the initial clustering error, making it an essential component for achieving an accurate low-rank approximation.

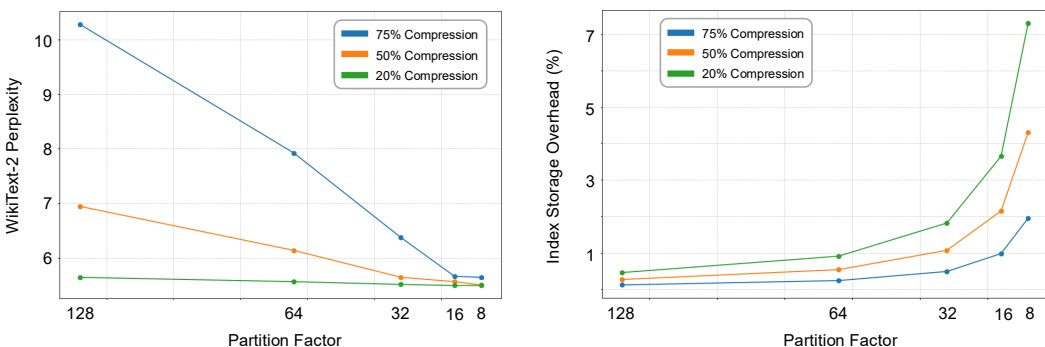

Figure 5: (Left) LLaMA2-7B model perplexity on varying row partition factor and compression ratios. (Right) LLaMA2-7B index storage overhead relative to the dense model size on varying row partition factor.

## A.4 VARYING ROW PARTITION FACTOR SIZES AND INDEX STORAGE OVERHEAD

We conduct an ablation study to analyze the impact of the row partition factor size on compression performance and index storage overhead. Figure 5 plots the perplexity at various compression ratios for partition sizes ranging from 128 down to 8. A clear trend emerges from the results: smaller partition sizes consistently yield better performance across all compression ratios.

This effect is most noticeable at the 75% compression ratio, where reducing the partition size from 128 to 16 decreases perplexity from 10.28 to 5.71. This suggests that a more fine-grained partitioning allows the clustering algorithm to find a more accurate representation of the weight subgroups. However, this performance gain comes with a trade-off in storage. Smaller partition sizes increase the number of groups, which in turn increases the memory required for the cluster indices. For example, at the same 75% compression ratio, reducing the partition size from 128 to 16 increases the index overhead from 0.12% to 1.95% of the original weight matrix size.

We also observe that the performance gains begin to plateau at smaller partition sizes. For instance, the difference in perplexity between a partition size of 16 and 8 is minimal but index storage almost doubles. Based on these results, we select a partition size of 16 for our main experiments, as it provides an excellent balance between high performance and minimal index storage overhead.

## A.5 ANALYSIS OF LEARNED VS. PREDEFINED CODEBOOKS

Table 5: Comparison with QuIP#

| Method | Effective bits | Codebook Entry # | LLaMA1 7B | LLaMA1 13B | LLaMA2 7B | LLaMA2 13B |
|---|---|---|---|---|---|---|
| Dense | 16 | - | 5.68 | 5.09 | 5.47 | 4.88 |
| QuIP# | 4 | $2^{32}$ | 5.76 | 5.17 | 5.56 | 4.95 |
| RoCCo (75%) | 4.16 | $2^{18}$ | 5.89 | 5.27 | 5.71 | 5.04 |

To better situate our work, we provide a direct comparison with QuIP# (Tseng et al., 2024), a state-of-the-art method that also uses a Hadamard transform followed by Vector Quantization (VQ). This comparison is insightful as it highlights the trade-offs between two different philosophies for VQ codebook design: a predefined, fixed codebook versus our learned, data-driven approach. The effective bit is calculated as $\frac{comp.modelsize * orig.bitwidth}{orig.modelsize}$

The fundamental difference between QuIP# and RoCCo lies in the nature of the codebook. QuIP# maps the transformed weights to a predefined, fixed codebook derived from the highly structured E8 lattice. This allows them to generate a massive number of virtual codebook entries ($2^{32}$) from a very

small (1KiB) low-precision source codebook, which can be viewed as having a very high number of quantization points but at a lower "codebook resolution." In contrast, RoCCo uses a data-driven, learned codebook. We apply k-means clustering and a subsequent Hessian-based update process to learn and optimize the centroids based on the specific distribution of the transformed weights. This results in fewer total codebook entries ($2^{18}$ for LLaMA2-7b QKV), but each centroid is stored at a higher precision, giving our method a higher "codebook resolution."

