# OpenReview forum: "RoCCo: Rotation-Augmented Clustering-based Low-rank Approximation for Compressing Large Language Models"
_ICLR.cc/2026/Conference — ICLR 2026 Conference Withdrawn Submission_

### Official Review · Reviewer_gH2n · 2025-10-29

**Soundness:** 2
**Presentation:** 1
**Contribution:** 1
**Rating:** 2
**Confidence:** 4

**Summary:**

This paper proposes the RoCCo (Rotation-augmented Clustering for Compression) technique for effectively compressing LLMs. The proposed technique aims to improve the accuracy of the compressed model using the Randomized Hadamard Transformation (RHT), which has been frequently discussed in recent quantization studies like QuIP# and QuaRot. The authors use Llama2 and OPT models to show the superiority of RoCCo.

**Strengths:**

RoCCo's approach to applying RHT for low-rank approximation is reasonable. Furthermore, RoCCo shows a significant performance gap with competitors by demonstrating its high accuracy even at very high compression ratios.

**Weaknesses:**

First, there are too many formatting errors throughout the paper. Table 2 extends beyond the page margins, the text in Figures 2 and 3 is difficult to read, and it is unclear why Section 5 is placed after the Experiments section.
Furthermore, all experiments are conducted on OPT and LLaMA2, which are very old models that are no longer in common use, so experiments on more recent models are necessary. Additionally, results from other benchmarks, such as GSM8K, are also needed.

**Questions:**

1. Could you provide experimental results for various models beyond Llama2, such as Llama 3 or Qwen?
2. Could you provide experimental results for GSM8K or other considerable benchmarks?

---

### Official Review · Reviewer_ovGL · 2025-11-01

**Soundness:** 2
**Presentation:** 3
**Contribution:** 2
**Rating:** 4
**Confidence:** 4

**Summary:**

The authors propose ROCCO, a compression pipeline for large language models consisting of three main steps: weight preconditioning using a random Hadamard transform to improve clustering, a divide-and-cluster procedure that finds representative cluster vectors in each submatrix, and a GPTQ-like second-order optimization step to minimize the layer output reconstruction error.

The approach is compared against several existing low-rank approximation methods. Additionally, the paper introduces an efficient attention mechanism compatible with rocco-compressed layers to further speed up inference.

**Strengths:**

- Strong empirical results compared to presented baselines.
- Creative adaptation of GPTQ-style calibration for low-rank approximation rather than quantization error.
- The “Efficient Attention Mechanism” section presents a practical improvement for faster computation in ROCCO-compressed attention layers.

**Weaknesses:**

- The novelty is limited and somewhat unclear.
- There is no comparison of inference speed with other low-rank methods.
- Compression time is not reported — important given the potential slowness of k-means clustering.
- The impact of the Hadamard transform beyond reconstruction error is not clearly justified; the consern
is that GPTQ-calibration you perform after clusterization makes the Hadamard preconditioning unnecessary.
- Some minor editing and clarity issues remain throughout the text.

Overall, the paper presents convincing empirical results on solid baselines. However, the contribution’s originality is ambiguous. It is unclear whether the clustering-based low-rank approximation itself is novel or if the authors mainly adapt existing methods. The necessity and contribution of the Hadamard preconditioning step are also questionable when GPTQ-style calibration is applied afterward.

**Questions:**

- Can you provide comparisons with clustering-based low-rank approximation + GPTQ centroid correction (without Hadamard transform) as an ablation study to isolate the effect of the rotation?
- Please clarify how LayerNorm and residual connections are handled when absorbing orthogonal matrices into adjacent weight matrices (lines 137–139).
- You do not cite prior clustering-based low-rank approximation methods, yet you also do not explicitly claim it as your contribution. Please clarify this point.
- Why do you omit ASVD and FWSVD from your comparisons, despite mentioning them in the related work?
- Provide inference time comparisons with other low-rank methods. As it stands, Tables 1 and 2 seem to exclude the efficient attention mechanism from Section 5, and Section 3.3 only provides complexity analysis relative to dense models.
- Please report the average compression time for ROCCO.
- Do you apply ROCCO to all linear layers in the model?
- The dense baseline should be visually separated from the compressed variants in the tables for readability.
- Please specify the dataset and metric used in Table 3.

---

### Official Review · Reviewer_taHP · 2025-11-04

**Soundness:** 2
**Presentation:** 2
**Contribution:** 1
**Rating:** 4
**Confidence:** 4

**Summary:**

This paper proposes a compression method for LLMs that combines orthogonal weight transformation via a randomized Hadamard transformation, clustering-based low-rank factorization, and clustered-attention computation for inference speedups. The goal of this work is to reduce model size without major accuracy loss, achieving high compression ratios while maintaining strong zero-shot performance.

The key idea is that LLM weight matrices contain vector-level outliers that distort clustering. The proposed method, RoCCo, first applies a random Hadamard transform to weight matrices to smooth the weight geometry (i.e., reduce vector-level outlier weights) to make the cluster centers more homogeneous. Subsequently, the weight matrix rows are split into groups, and k-means clustering per group is performed to get centroids and assignment indices. Then, the centroids are refined using a Hessian-based update to reduce layer output error. Optionally, core attention compute can be done directly in the centroid space, reducing FLOPs.

Experiments on Llama and OPT models show that it outperforms prior approaches on standard zero-shot accuracy evaluations and Wikitext-2 PPL. Furthermore, low-rank approximation can be combined with GPTQ to produce usable models even in extremely low-bit settings.

**Strengths:**

The paper studies the important problem of LLM compressing, focusing on structural approaches like low-rank approximation. The insight behind the paper is well-motivated -- orthogonal transformations indeed reduce the vector-level outliers that distort clustering, as can be seen in Fig. 1.

Amongst the numerical evaluations done in this paper, RoCCo preserves accuracy on Llama2 and OPT models, demonstrating practical significance. Furthermore, by showing that RoCCo integrates cleanly with GPTQ and enables usable models at $\sim 2$ bits effective precision (where GPTQ alone fails), the paper establishes orthogonality to quantization while highlighting a potential compression strategy that utilizes the best of both world.

Finally, the clustered-attention mechanism extends the compression framework to algorithmic speed-ups, reducing FLOPs on small model. While not the main driver of results, it shows potential for deployment efficiency.

**Weaknesses:**

One of the major concerns I have is that the numerical evaluations are pretty limited (i.e., only older Llama-2 and OPT models have been evaluated). It is well known that more recent models like the Llama-3 family, Qwen, etc. are less amenable to quantization, and results of the newer models which are more commonly used nowadays is important to convince the usefulness of the approach.

Moreover, existing works seems to already use Hadamard transformation with low-rank approximation + quantization. For example, in [this paper](https://proceedings.neurips.cc/paper_files/paper/2024/file/a20e8451ffb07ad25282c21945ad4f19-Paper-Conference.pdf), random Hadamard transform is applied to the left and right of the input weight matrix as the incoherence pre-processing step and it helps both quantization (using QuIP#) and low-rank approximation. Discussions of howRoCCO compares with prior works such as this should be included.

Finally, even though the authors show how clustered-attention can lead to FLOPs reduction, throughput numbers (e.g., time-to-first-token or inter-token-latency) are not reported. Often, FLOP reduction does not lead to a latency reduction, especially in the memory-bound auto-regressive decoding phase, where most of the time is spend in data transfers and compute units stay idle. If possible, results  with actual numbers should be included, and if not this limitation should be explicitly acknowledged.

**Questions:**

1. The authors say in line 088, *Unlike element-wise methods that can produce sparse matrix representations, LRA-based methods typically result in smaller, dense matrices, which have high efficiency on modern hardware like GPUs.* This is not always true. Recent accelerators (e.g., TPUs) maximize compute utilization when the matrices are large. Reducing the size of the matrices just leads to wasted compute (low utilization), with potential accuracy loss.

2.   Line 103. ModeGPT .... is incomplete.

3. It is stated in line 151, *While the weights contain significant structural redundancy due to the structured computational graph of transformers (eg. Multi-head self-attention) (Vaswani et al., 2017), their distributions are often characterized by the existence of outliers.* This seems like a hand-wavy justification -- please provide rigorous justification if possible. If not, simply the empirical observation that weight matrices are low-rank should suffice.

---

### Official Review · Reviewer_hf7x · 2025-11-04

**Soundness:** 2
**Presentation:** 2
**Contribution:** 3
**Rating:** 4
**Confidence:** 3

**Summary:**

This paper proposes a large language model (LLM) compression algorithm that integrates rotational preconditioning with clustering-based optimization. On small- and medium-scale models, the method achieves state-of-the-art compression performance, significantly outperforming traditional matrix decomposition–based approaches.

**Strengths:**

1.	The proposed method outperforms other structured pruning approaches by a large margin, as demonstrated in Table 2.
2.	Rotational transformations, previously shown to be effective in quantization, are further validated in this paper to be equally beneficial for pruning, highlighting their general applicability across different compression paradigms.

**Weaknesses:**

1.	The quantitative analysis in the ablation studies is insufficient; it remains unclear how each component—rotation and clustering—contributes individually to the final performance.
2.	The experimental evaluation lacks large-scale model exploration, limiting the understanding of the method’s scalability.
3.	The inference efficiency of the compressed model is not analyzed, leaving its practical runtime benefits uncertain.
4.	The comparison of compression time across different methods is missing, making it difficult to assess the computational cost of the proposed approach.

**Questions:**

Could you provide additional experiments to address the weaknesses mentioned above?
Also, can your clustering method be formulated as a type of decomposition approach?

---

### Official Review · Reviewer_9jdF · 2025-11-07

**Soundness:** 1
**Presentation:** 3
**Contribution:** 2
**Rating:** 4
**Confidence:** 5

**Summary:**

This paper introduces RoCCo, a post-training model compression method. The core technique involves using a fast Walsh–Hadamard transform (FWHT) to remove outliers in weight principal component analysis (PCA). Subsequently, a clustering method like K-means replaces weight columns with their corresponding cluster centroids. The paper also proposes a novel approach to bypass the reconstruction of the large weight matrix, enabling direct attention calculation using the grouped clusters and activations.

**Strengths:**

- The paper presents an interesting observation on how the rotation (specifically, the randomized Hadamard transform - RHT) aids clustering for weight compression.
- It achieves an impressive performance increase in zero-shot accuracy, even with a 75% compression rate.

**Weaknesses:**

- The effectiveness of the rotation is only empirically observed, lacking mathematical proof. The authors do not convincingly demonstrate the benefit of applying an RHT to the weight matrix, especially since the resulting PCA figure appears scattered.
- The method's concept of approximating the original weights with smaller storage without reconstruction is analogous to quantization. Therefore, the paper should be compared against quantization methods, not just other post-training compression techniques.

**Questions:**

- What is the actual speedup of this method? While the fast RHT is noted as $O(n \log n)$, how does the calibration training speed compare to other post-training compression methods?
- Since other compression methods can be combined with quantization, how does RoCCo's performance compare to these combined approaches?

---

### Note · Authors · 2025-12-02

I have read and agree with the venue's withdrawal policy on behalf of myself and my co-authors.